# Characterizing the Bone Marrow Environment in Advanced-Stage Myelofibrosis during Ruxolitinib Treatment Using PET/CT and MRI: A Pilot Study

Stefanie Slot [1,*], Cristina Lavini [2], Gerben J. C. Zwezerijnen [3], Bouke J. H. Boden [4], J. Tim Marcus [3], Marc C. Huisman [3], Maqsood Yaqub [3], Ellis Barbé [5], Mariëlle J. Wondergem [1], Josée M. Zijlstra [1], Sonja Zweegman [1] and Pieter G. Raijmakers [3]

1 Department of Hematology, Amsterdam UMC Location Vrije Universiteit Amsterdam, Cancer Center Amsterdam, De Boelelaan 1117, 1081 HV Amsterdam, The Netherlands
2 Department of Radiology and Nuclear Medicine, Amsterdam UMC Location University of Amsterdam, Meibergdreef 9, 1105 AZ Amsterdam, The Netherlands
3 Department of Radiology and Nuclear Medicine, Amsterdam UMC Location Vrije Universiteit Amsterdam, De Boelelaan 1117, 1081 HV Amsterdam, The Netherlands
4 Department of Radiology, Onze Lieve Vrouwe Gasthuis, Oosterpark 9, 1091 AC Amsterdam, The Netherlands
5 Department of Pathology, Amsterdam UMC Location Vrije Universiteit Amsterdam, De Boelelaan 1117, 1081 HV Amsterdam, The Netherlands
* Correspondence: s.slot1@amsterdamumc.nl

**Abstract:** Current diagnostic criteria for myelofibrosis are largely based on bone marrow (BM) biopsy results. However, these have several limitations, including sampling errors. Explorative studies have indicated that imaging might form an alternative for the evaluation of disease activity, but the heterogeneity in BM abnormalities complicates the choice for the optimal technique. In our prospective diagnostic pilot study, we aimed to visualize all BM abnormalities in myelofibrosis before and during ruxolitinib treatment using both PET/CT and MRI. A random sample of patients was scheduled for examinations at baseline and after 6 and 18 months of treatment, including clinical and laboratory examinations, BM biopsies, MRI (T1-weighted, Dixon, dynamic contrast-enhanced (DCE)) and PET/CT ([15O]water, [18F]NaF)). At baseline, all patients showed low BM fat content (indicated by T1-weighted MRI and Dixon), increased BM blood flow (as measured by [15O]water PET/CT), and increased osteoblastic activity (reflected by increased skeletal [18F]NaF uptake). One patient died after the baseline evaluation. In the others, BM fat content increased to various degrees during treatment. Normalization of BM blood flow (as reflected by [15O]water PET/CT and DCE-MRI) occurred in one patient, who also showed the fastest clinical response. Vertebral [18F]NaF uptake remained stable in all patients. In evaluable cases, histopathological parameters were not accurately reflected by imaging results. A case of sampling error was suspected. We conclude that imaging results can provide information on functional processes and disease distribution throughout the BM. Differences in early treatment responses were especially reflected by T1-weighted MRI. Limitations in the gold standard hampered the evaluation of diagnostic accuracy.

**Keywords:** myelofibrosis; MRI; PET/CT; diagnostic accuracy; ruxolitinib

## 1. Introduction

Primary myelofibrosis (MF) is a relatively rare disease that belongs to the group of myeloproliferative neoplasms (MPNs). The disease mainly affects the elderly, with a median age at diagnosis of 67 years [1]. Apart from constitutional symptoms and splenomegaly, MF patients show various bone marrow (BM) alterations, caused by driving mutations and increased cytokine production [2–4]. Typically, early MF is characterized by myeloid hypercellularity and abundant atypical megakaryocytes, whilst fibrosis and osteosclerosis

predominate in later stages [5], often accompanied by neoangiogenesis [6,7]. Median survival in MF varies between 16 and 185 months, and prognosis is commonly estimated using the (dynamic) International Prognostic Scoring System ((D)IPSS) or the mutation-enhanced IPSS (MIPSS) scores [8–10]. Treatment in MF is largely symptomatic, but several promising drugs have been developed following the discovery of driver mutations, including the JAK1/2 inhibitor ruxolitinib. Interestingly, ruxolitinib often reduces clinical symptoms within the first 12 weeks of treatment [11,12], but it was only after two and four years that fibrosis reduction was seen in 15% and 30% of patients treated in clinical trials, respectively [12,13]. Amongst others, this questions whether current diagnostic methods are sensitive enough to detect early BM changes during treatment.

Indeed, the current hallmark of diagnostic and treatment response criteria—the BM biopsy [14–16]—has several limitations. First, hard bone structures and crushing artifacts can lead to inadequate BM samples. Moreover, inhomogeneous disease distribution can cause "sampling error" [17]. Even in assessable samples, interobserver variability in fibrosis grading can be high [13,15] and small differences in fibrosis may be missed due to coarseness in the grading system. Moreover, information on functional processes, such as osteoblast activity and blood flow, is limited. Lastly, BM biopsies are invasive and, although rare, may cause serious bleeding complications [18]. Given these limitations, an alternative diagnostic method is desirable.

Although not currently used in clinical practice, imaging might form an alternative for the evaluation of MF disease activity. Based on the limited literature [19], several techniques seem particularly interesting. T1-weighted magnetic resonance imaging (MRI) and Dixon imaging have been shown to visualize BM fat content in MF [20,21] and thereby might indirectly reflect BM cellularity and/or fibrosis [17]. For the evaluation of osteosclerosis, conventional radiography is insufficiently accurate, but PET or scintigraphy combined with a bone-seeking tracer (e.g., [18F]sodium fluoride ([18F]NaF) might visualize osteoblastic activity [22–25]. Lastly, BM blood flow in MF has sporadically been evaluated via dynamic contrast-enhanced MRI (DCE-MRI) and PET [22,26–30]. Although the optimal tracer for this purpose is unknown, [15O]water PET is the established gold standard for quantification of blood flow in vivo in various other organs [31].

Whilst previous explorative studies have indicated a possible use for the above-named diagnostic imaging techniques in MF, diagnostic accuracy was seldom evaluated. Moreover, most studies focused on one imaging technique, generally reflecting only one biological aspect of the disease (i.e., cellularity, vascularization, etc.,). To our knowledge, only one study evaluated the effect of ruxolitinib on the BM imaging appearance [21]. In our current study, we aimed to visualize the whole spectrum of BM abnormalities in MF before and during ruxolitinib treatment, using a combined imaging protocol with a direct comparison to histopathological results.

## 2. Materials and Methods

### 2.1. Study Aim

The primary objective was to compare imaging results with histopathological BM characteristics, regarding fatty, cellular, and fibrotic components, bone new formation, and vascularization. Secondary objectives were to explore the value of different imaging techniques in ruxolitinib response monitoring in MF, as well as the occurrence of BM biopsy sampling errors.

### 2.2. Study Design and Patient Selection

In this prospective diagnostic pilot study, we included a random sample of MF patients who were scheduled for ruxolitinib treatment and who met the following inclusion criteria: A diagnosis of primary MF according to the WHO 2008 criteria [14]; grade 3 BM fibrosis; and an intermediate-1, -2 or high risk according to the IPSS [8]. Patients were ineligible for participation in case of current or previous treatment with a JAK2 inhibitor, a previous allogeneic stem cell transplantation (allo-SCT), a contraindication for used imaging techniques,

and inability to sign informed consent. Patients were recruited via treating physicians in a large teaching hospital and an academic hospital in the Netherlands. Central medical ethics committee approval was obtained. In addition, we obtained control values of nine persons without MF for comparisons. These patients had participated in previous trials, and had signed informed consent for study participation and/or (re-)use of data.

*2.3. Study Protocol*

2.3.1. Study Procedures

Ruxolitinib prescription and (dis)continuation took place by the treating physician according to standard care. Study investigations were done in the academic hospital in the month before start of treatment (T0), after six months (T6), and after 18 months (T18). The timing of these follow-up evaluations was based on the likelihood of clinical changes and histopathological changes to have occurred, respectively. Investigations consisted of a patient history, physical examination, the Myeloproliferative Neoplasm Symptom Assessment Form (MPN-SAF) [32], blood laboratory tests, a BM biopsy, and imaging according to the study protocol (Table 1). For the BM biopsy, as many attempts as needed were made to obtain a sample of >1 cm in length (preferably > 2 cm). Adverse events were scored according to the NCI Common Terminology Criteria for Adverse Events, version 3.0.

**Table 1.** Imaging protocol.

| Technique | Regions of Interest [a] | Outcome Measure(s) |
|---|---|---|
| T1-weighted MRI | Spinal column, pelvis, proximal femora | Visual signal intensity (low/normal/high, homo- or heterogeneous distribution) [b] |
| Dixon | Th 5/7/9, L1/4, pelvis, proximal femora | Fat–water signal ratio |
| DCE-MRI | Th 7/9, L1 | $K^{trans}$, $V_e$, $K_{ep}$ |
| 15O-water PET | Th 5/7/9 | Blood flow in ml/min/ml |
| 18F-NaF PET | Th 5/7/9, L1/4, pelvis, proximal femora | Ki, SUVmean, SUVintegral |

(DCE-)MRI = (dynamic contrast-enhanced) magnetic resonance imaging, $K^{trans}$ = volume transfer constant between plasma and EES in $min^{-1}$ (representing blood plasma flow or permeability surface area product per unit volume of tissue depending on permeability conditions), ME = maximum contrast-enhancement (in arbitrary units), PET = positron emission tomography, SUV = standardized uptake value, $V_e$ = volume of extravascular extracellular space (EES) per unit volume of tissue. [a] Regions of interest were drawn in the medullary cavity of selected vertebrae, the proximal femora at both the level of the femoral head and the subtrochanteric shaft, and the posterior iliac bone adjacent to the sacroiliac joint (left side only for Dixon, bilaterally for [18F]NaF PET).[b] Comparison with adjacent muscle and/or intervertebral disk.

2.3.2. Conventional Evaluation

Risk score at T0 was defined according to the DIPSS-plus, which was the standard risk assessment tool at the time [9]. Response was graded according to the 2013 IWG-MRT/ELN criteria, which includes the MPN-SAF total symptom score (TSS) [16,32]. Iliac crest BM samples were assessed by a dedicated pathologist from the Dutch MPN pathology expert panel. Fibrosis severity and age-adjusted cellularity were evaluated according to the 2005 European Consensus Criteria [33]. Fat content was visually graded on a 3-point scale (low/normal/high). Microvessel density (MVD) was quantified as the median number of microvessels per high-power field as identified by CD34+ staining [6]. Osteosclerosis severity was assessed according to recent consensus criteria [34].

2.3.3. Imaging Protocol

A summary of the imaging protocol including regions of interest (ROIs) and outcome measures is provided in Table 1.

A PET/MR scanner (Philips Ingenuity TF, 3.0 Tesla) was used for MR imaging. A spine coil was connected for sagittal spinal column images. Whole-body (skull base-midthigh) T1-weighted spin-echo was performed in the coronal plane (four slabs with repetition time (TR) 500–750 ms, echo time (TE) 17.5 ms, matrix size 208 × 280, 36 slices of 6 mm

per slab). A whole-body Dixon sequence was performed in the axial plane (eight slabs of matrix size $300 \times 200$, 17 slices of 8 mm per slab), including T1-in phase, T1-opposed phase, water, and fat images. DCE-MRI was performed in the sagittal plane using a 3D T1-weighted (RF spoiled) Fast Field Echo sequence with the field of view (FOV) covering the thoracolumbar spine using the following parameters: TR/TE 3.9/1.58 ms, flip angle 12 deg, FOV 320 mm $\times$ 320 mm, matrix size $180 \times 147$, in plane resolution 1.25 mm, 5 slices of 4.5 mm, 70 dynamic scans with a temporal resolution of 4.91 s. The DCE-MRI images were acquired before, during and after the intravenous administration of a gadolinium-based contrast agent (Dotarem 0.2 mL/Kg) with standardized injection speed (3 mL/s). In addition, a set of separate fast T1-weighted sequences (with the same scanning parameters as those of the DCE-MRI scans) using five different values of the flip angle (2, 5, 10, 12 15 deg) were acquired before the contrast agent was injected, in order to calculate the native T1 relaxation time of the tissues using the Variable Flip Angle (VFA) method [35]. Non-linear fitting was used to obtain the T1. Contrast agent concentrations were obtained from the dynamic signal enhancement and the pre-contrast T1 values using Equation (1), where TR is the repetition time, $T_{10}$ is the pre-contrast T1 value as measured with the VFA method, alpha is the flip angle, r1 is the contrast agent relaxivity, and $S_{10}(t)$ is the signal ratio between the signal at time(t) and the pre-contrast signal.

$$C_{Gd}(t) = -\frac{1}{TR \cdot r_1} \ln \left\{ \frac{S_{10}(t) \cdot \left(1 - e^{-\frac{TR}{T_{10}}}\right) - \left(1 - \cos\alpha \cdot e^{-\frac{TR}{T_{10}}}\right)}{e^{-\frac{TR}{T_{10}}} \left(S_{10}(t) \cdot \cos\alpha \cdot \left(1 - e^{-\frac{TR}{T_{10}}}\right) - \left(1 - \cos\alpha \cdot e^{-\frac{TR}{T_{10}}}\right)\right)} \right\} \quad (1)$$

PET scanning was performed on a PET/CT scanner (Philips Ingenuity TF), with a total acquisition time of approximately 90 min. After acquisition of a topogram of the FOV (80 kV, 20 mAs, thoracic aorta upper margin-left ventricle lower margin), a standard dose of 370MBq [$^{15}$O]water was administered via the antecubital vein. The dynamic protocol consisted of 26 frames in 10 min ($1 \times 10$ s, $8 \times 5$ s, $4 \times 10$ s, $2 \times 15$ s, $3 \times 20$ s, $2 \times 30$ s, $6 \times 60$ s). Finally, a low-dose CT (120 kV, 30 mAs) was obtained. After acquisition of another topogram of the FOV (thoracic aorta upper margin-left ventricle lower margin), 2.1 MBq [$^{18}$F]NaF/kg body weight was administered intravenously. For the dynamic protocol, a total of 26 frames was recorded in 45 min ($1 \times 10$ s, $4 \times 5$ s, $3 \times 10$ s, $3 \times 20$ s, $2 \times 30$ s, $7 \times 60$ s, $2 \times 150$ s, $2 \times 300$ s, $2 \times 600$ s) and venous blood samples were drawn at 20, 30, and 40 min scanning time for calibration of image-derived input function. The static [$^{18}$F]NaF PET scan extended from the jugular notch to the groin and scanning was performed with 3 min bed positions. Low-dose CT was used for attenuation correction.

For all scans, the first author determined the placement of ROIs. The analysis of MRI results was done by a team of three operators, who were not involved in reading of PET/CT and/or histopathological results. T1 and Dixon images were analyzed by a dedicated radiologist, using endpoints as specified in Table 1. DCE-MRI data were quantitatively analyzed by two dedicated physicists with the Tofts model [36], using a population-averaged arterial input function (AIF) and in-house developed software [37]. For the quantitative analysis, we calculated previously described outcome measures [36,38] and produced maps of the pharmacokinetic parameters $K^{trans}$, $V_e$, and $K_{ep}$. The analysis of PET/CT data was performed by a team of four operators (who were not involved in reading of MRI and/or histopathological results), including two dedicated nuclear medicine physicians and two physicists. For absolute quantification of [$^{18}$F]NaF uptake, image-derived input functions from the thoracic aorta and tissue [$^{18}$F]NaF activity measurements from the dynamic PET scan were used in a two-compartment kinetic model, yielding the Ki. This parameter represents the absolute net clearance of [$^{18}$F]NaF from plasma to bone [25]. Furthermore, we obtained standardized uptake values (SUV) using in-house developed Matlab-based software. For the choice of the optimal SUV, we compared the absolute value Ki with both SUVmean values and SUVintegral values using simple linear regression.

### 3. Results

#### 3.1. Study Population

Four patients were enrolled in the study (3 males/1 female, aged 47–75 years), of whom three completed the entire study protocol. Pt (patient) 2 died before T6, due to transformation to acute myeloid leukemia. Patient, disease, and treatment characteristics are listed in Table 2. The control patient for DCE-MRI was a 66-year old male patient who had previously undergone back surgery because of herniated disks (at level Th5-6) and lumbar spinal stenosis (at level L4-5). Control patients for [$^{15}$O]water PET included 6 males and 2 females, aged 42–87 years, who were evaluated for coronary artery disease.

**Table 2.** Patient, disease, and treatment characteristics.

|  | **Pt 1** | **Pt 2** | **Pt 3** | **Pt 4** |
|---|---|---|---|---|
| Sex | male | female | male | male |
| Age at T0, years | 75 | 64 | 47 | 65 |
| Time since diagnosis, months | 31 | 2 | 30 | 262 |
| Driving mutation | JAK2V617F | JAK2V617F | JAK2V617F | calreticulin |
| Prior treatment | - | - | - | - |
| Comorbidities | - | atrial fibrillation, hypertension | - | - |
| Transfusion history | no | yes (2 months) | no | no |
| DIPSS plus risk score at T0 | int-1 | int-2 | int-1 | int-2 |
| Main reason for treatment | abd. discomfort | night sweats, itching | night sweats, abd. discomfort | night sweats, weight loss |
| Ruxolitinib starting dose, mg/day | 40 | 40 | 40 | 40 |
| Dose adjustments (reason) | no | Dose decrease to 5 mg/day (cytopenia) | no | Dose decrease to 20 mg/day (anemia) |
| Grade ≥ 3 adverse events | no | neutropenia | no | anemia |
| Blood transfusions since T0 | no | yes | no | yes |
| Follow-up | deceased | transformation AML | alive | alive |

Abd. = abdominal, AML = acute myeloid leukemia, (D)IPSS = (dynamic) international prognostic scoring system, MF = myelofibrosis, Pt = patient

#### 3.2. Conventional Response Evaluations

A summary of histopathological results and clinical evaluations for individual patients is given in Table 3. Of note, the final visit for Pt 1 was postponed because of family matters and was termed T24.

All patients showed grade 3 fibrosis with osteosclerosis at baseline, and no significant reductions in fibrosis grade were observed during treatment. Of note, BM biopsies were not evaluable in two cases due to crushing artifacts. Despite the absence of major histopathological changes, a significant splenic response and a decrease in physician-reported disease-related symptoms was noted in all three evaluable patients. Moreover, a significant decrease in MPN-SAF TSS was found in one patient (Pt4). According to the IWG-MRT/ELN criteria—which combined the above named and additional items—two patients obtained clinical improvement (Pt1 and Pt3). Pt 4 did experience a decrease in disease-related symptoms, but required dose reductions because of progressive thrombocytopenia and was therefore classified as having stable disease.

**Table 3.** Conventional response evaluation.

| | Spleen Volume * | MPN-SAF TSS | Bone Marrow Biopsy | | | | IWG-MRT/ELN ^ | DIPSS Plus |
| | | | Cellularity | Fibrosis ^ | Fat | MVD | | |
|---|---|---|---|---|---|---|---|---|
| Pt 1 | | | | | | | | |
| T0 | 3580 mL | 19 | high | 3 (+) | low | n.e. | - | int-1 |
| T6 | 1970 mL | 22 | low | 2-3 (+) | ↑ | 23/HPF | CI | int-1 |
| T24 | 1600 mL | 23 | n.e. | n.e. (+) | n.e. | n.e. | CI | int-2 |
| Pt 2 | | | | | | | | |
| T0 | 480 mL | unknown | high | 3 (+) | low | 12/HPF | - | int-2 |
| Pt 3 | | | | | | | | |
| T0 | 2920 mL | 14 | high | 3 (+) | low | 22/HPF | - | int-1 |
| T6 | 2140 mL | 14 | normal | 3 (+) | ↑ | 33/HPF | SD | int-1 |
| T18 | 1780 mL | 28 | high | 3 (+) | Low (↓) | 32/HPF | CI | int-1 |
| Pt 4 | | | | | | | | |
| T0 | 2490 mL | 46 | low | 3 (+) | low | 33/HPF | - | int-2 |
| T6 | 1830 mL | 26 | low | 3 (+) | ↑ | 26/HPF | SD | int-2 |
| T18 | 1520 mL | 22 | n.e. | n.e. (+) | n.e. | n.e. | SD | int-2 |

* As measured on MRI: 30 + 0.58 × maximum caudocranial dimension × maximum size × thickness (measured in centimeters in the axial plane). ^ The fibrosis grade is followed by an indication of the presence or absence of osteosclerosis (in brackets). CI = clinical improvement, int = intermediate, MVD = microvessel density, Pt = patient, SD = stable disease, n.e. = not evaluable due to severe crushing artifacts. ↑ = increase from previous measurement, ↓ = decrease from previous measurement.

### 3.3. Imaging Results

Imaging results are graphically shown in subsequent paragraphs. A comparison of imaging results per patient is given in Table 4. Of note, in patient 3 Th8 was used instead of Th9, because of a hemangioma in Th9.

**Table 4.** Imaging results.

| | MRI T1 [a] | | Dixon Fat–Water Signal Ratio [b] | | DCE-MRI [c] (Spine) | | $^{15}$O-Water PET Flow [d] (Spine) | $^{18}$F-NaF PET SUVintegral [e] | |
| | Pelvis | Spine | Pelvis | Spine | $V_e$ | $K^{trans}$ | | Pelvis | Spine |
|---|---|---|---|---|---|---|---|---|---|
| | | | | | Pt 1 | | | | |
| T0 | 1-2P | 1P | 0.69 (−) | 0.26 (−) | - | - | 0.33 (−) | 0.08 (−) | 0.08 (−) |
| T6 | 2P | 1P | 1.48 (+114%) | 0.32 (+23%) | +54% | −38% | 0.17 (−48%) | 0.05 (−37%) | 0.08 (=) |
| T24 | 2P | 1-2P | 3.37 (+388%) | 1.32 (+407%) | +90% | −30% | 0.23 (−30%) | 0.05 (−37%) | 0.07 (−12%) |
| | | | | | Pt 2 | | | | |
| T0 | 1H | 1H | 0.43 (−) | 0.26 (−) | - | - | 0.40 (−) | 0.12 (−) | 0.11 (−) |
| | | | | | Pt 3 | | | | |
| T0 | 1P | 1H | 0.5 (−) | 0.22 (−) | - | - | 0.37 (−) | 0.11 (−) | 0.09 (−) |
| T6 | 2P | 1P | 1.14 (+128%) | 0.28 (+27%) | +11% | +1% | 0.45 (+22%) | 0.12 (+9%) | 0.09 (=) |
| T18 | 2P | 1P | 1.12 (+124%) | 0.16 −27%) | −1% | −13% | 0.54 (+46%) | - | 0.08 (−11%) |
| | | | | | Pt 4 | | | | |
| T0 | 1H | 0H | 0.77 (−) | 0.34 (−) | - | - | 0.61 (−) | 0.14 (−) | 0.22 (−) |

**Table 4.** *Cont.*

| | MRI T1 [a] | | Dixon Fat–Water Signal Ratio [b] | | DCE-MRI [c] (Spine) | | ¹⁵O-Water PET Flow [d] (Spine) | ¹⁸F-NaF PET SUVintegral [e] | |
|---|---|---|---|---|---|---|---|---|---|
| | Pelvis | Spine | Pelvis | Spine | $V_e$ | $K^{trans}$ | | Pelvis | Spine |
| T6 | 1H | 0H | 1.44 (+87%) | 0.76 (+124%) | +33% | +34% | 0.63 (+3%) | 0.17 (+21%) | 0.23 (+5%) |
| T18 | 1H | 0H | 1.34 (+74%) | 0.48 (+41%) | +153% | +185% | 0.66 (+8%) | 0.13 (−7%) | 0.20 (−11%) |

[a] Results are presented on a 3-point scale according to the signal intensity compared with adjacent muscle or intervertebral disk (0 = hypointense, 1 = isointense, 2 = hyperintense). Additionally, the signal distribution is labeled as homogeneous (H) or patchy (P). [b] Results are presented as fat–water fractions and (percentage changes) from baseline. [c] Results are presented as percentage changes from baseline (absolute values at T0 are not presented). [d] Results are presented as ml/min/mL and (percentage changes) from baseline (thoracolumbar spine). [e] Results are presented as SUV integral values and (percentage changes) from baseline (thoracic spine).

### 3.3.1. Axial and Proximal Femoral T1-Weighted MRI

Abnormal (i.e., hypointense or isointense) vertebral, pelvic, and femoral signal intensities were seen in all patients at baseline (Figure 1). During treatment, two patients (Pt1 and Pt3) showed an increase in pelvic and femoral signal intensities at T6, followed by an increase in vertebral signal intensity at T18/T24. One patient (Pt4) showed stable signal intensities in all ROIs during treatment.

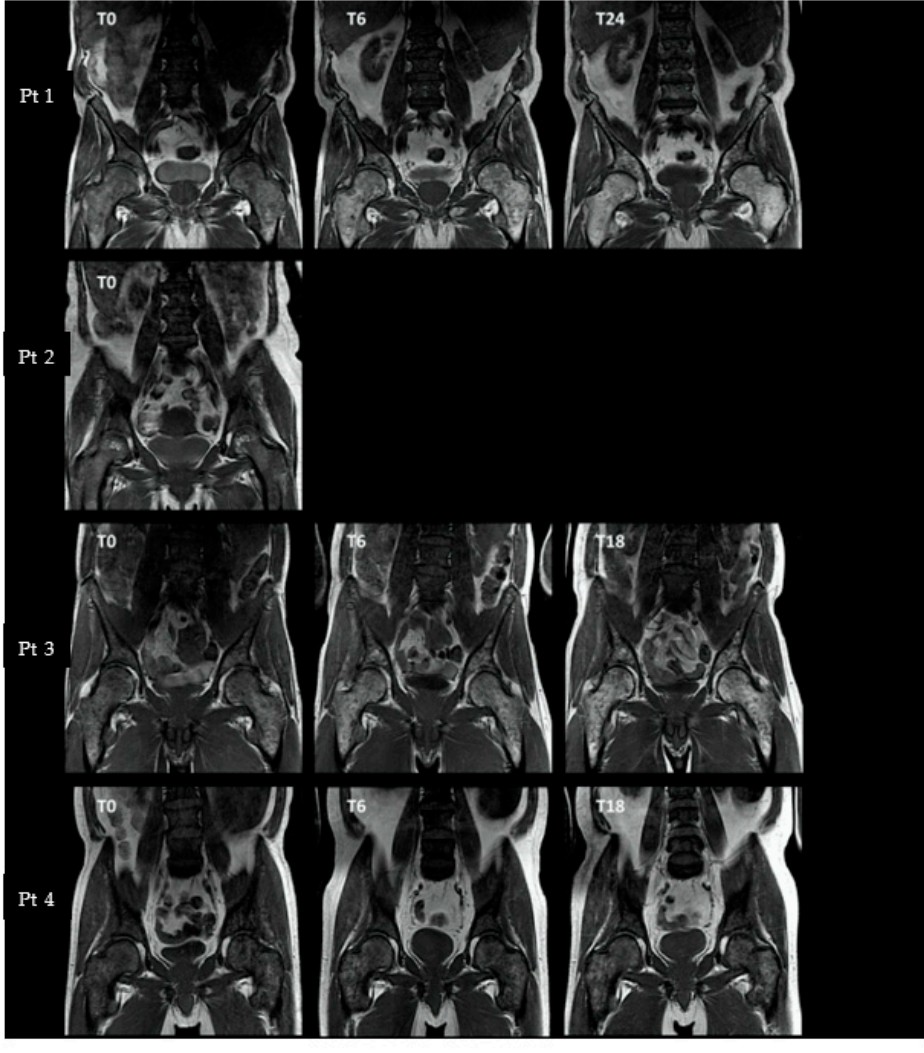

**Figure 1.** T1-weighted MRI images. Coronal T1-weighted images are shown for each individual

patient at available time points. T0 represents the MRI at baseline, T6 the MRI at 6 months, and T18 and T24 the MRI at 18 or 24 months, respectively. Pt 1 = patient 1 (etc.). T1-signal intensity was scored based on multiple slices, using methods as outlined above. Abnormal (i.e., isointense or hypointense) vertebral, pelvic, and femoral signal intensities were seen in all patients at baseline (T0). During treatment, patients 1 and 3 showed an increase in femoral and pelvic signal intensities at T6, followed by an increase in vertebral signal intensities at T18/24. Patient 4 showed overall stable signal intensities during treatment.

### 3.3.2. Axial and Proximal Femoral Dixon

In all four patients, baseline fat–water fractions were lowest in the vertebrae, followed by the pelvis and femoral regions (Figure 2). One patient (Pt1) showed an increase in fat–water fractions in all ROIs at both T6 and T24. The fastest increase occurred in the femora, followed by the pelvis and vertebrae, respectively. In one patient (Pt3), vertebral and femoral head fat–water fractions at T6 were stable, whilst pelvic and femoral shaft values increased slightly. At T18, femoral fat–water fractions increased further, whilst vertebral values decreased to below baseline and pelvic values remained stable. In one patient (Pt4), fat–water fractions in all ROIs increased at T6. At T18, a subsequent decrease in the vertebral fat–water fraction was seen, with stable pelvic and femoral values.

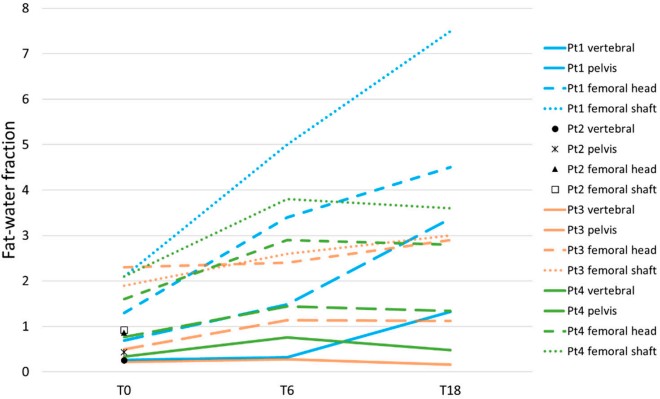

**Figure 2.** Fat–water fractions as measured by Dixon.

### 3.3.3. Vertebral DCE-MRI

An example of a $K^{trans}$ map is shown in Figure 3. In one patient (Pt 1), the $K^{trans}$ decreased at T6 and remained stable afterwards (Figure 4). In one patient (Pt 3), the $K^{trans}$ remained unchanged during treatment, whilst it increased both at T6 and T18 in Pt 4.

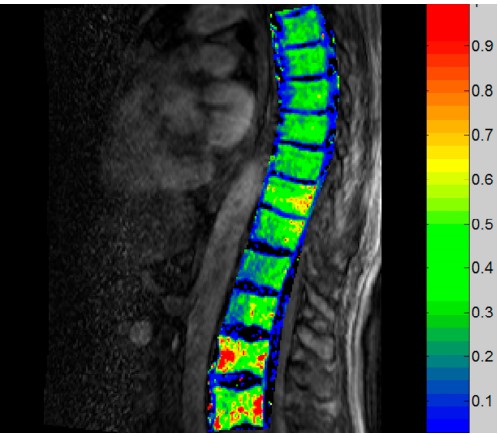

**Figure 3.** Example of $K^{trans}$ map. The figure represents the $K^{trans}$ map of Pt 1 at T0. The different colors correspond to $K^{trans}$ values in $min^{-1}$, as shown in the right.

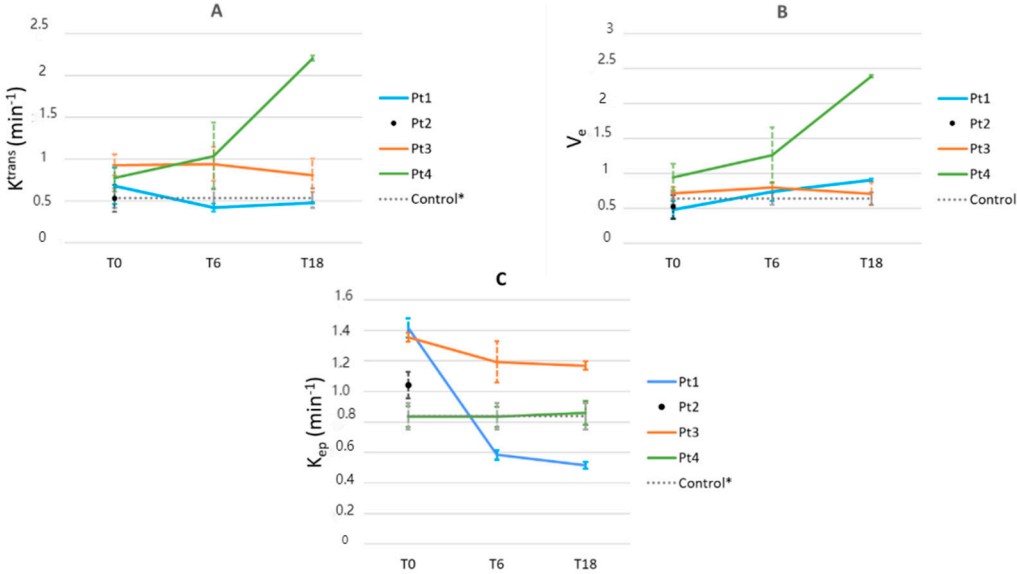

**Figure 4.** $K^{trans}$ and $V_e$ as measured by DCE-MRI. Mean values (+/−SD) for $K^{trans}$, $V_e$, and $K_{ep}$ across three vertebrae (Th7, Th9, and L1) of each individual patient are shown in panels (**A**–**C**), respectively. Vertical dashed lines represent the standard deviation of the mean of the measured values in the three individual vertebrae. NB: values of $V_e$ are > 1 in one patient. This is most probably a consequence of using a standard (thus not measured) Arterial Input function. * The healthy control was examined at baseline only; results were extrapolated for later time points.

Two patients (Pt1 and Pt4) showed an increase in $V_e$ at T6, with a further increase at T18/T24. One patient (Pt3) showed stable $V_e$ values during treatment.

Regarding $K_{ep}$, a significant decrease was seen at T6 in Pt 1. A minor decrease in $K_{ep}$ was found in Pt 3. In Pt 4, $K_{ep}$ values remained stable throughout treatment.

### 3.3.4. Vertebral Blood Flow as Measured by [$^{15}$O]Water PET

At baseline, all patients had an increased vertebral blood flow compared with normal control values (mean value 0.426 mL/min/mL vs. 0.236 mL/min/mL, $p < 0.05$) (Figure 5). Pt 1 showed a normalization of blood flow at T6. In Pt 3, blood flow increased further at T6 and T18. Pt 4 showed a persistently increased blood flow at all measurements.

### 3.3.5. Axial and Femoral [$^{18}$F]NaF Uptake

Comparison of SUVmean and SUVintegral with the gold standard Ki showed correlation r values of 0.55 and 0.99, respectively. Therefore, SUVintegral values are presented for the different predefined ROIs. In addition, we report baseline Ki for comparative interpretation of baseline measurements.

Mean vertebral Ki values ranged from 0.058–0.079 mL/min/mL in three patients (Pt 1–3, and an extreme value of 0.175 mL/min/mL was measured in Pt 4 (Figure 6). All three evaluable patients showed more or less stable vertebral SUVintegral values during treatment. In one patient (Pt1), femoral and pelvic SUVintegral values decreased at T6 and T24. In one patient (Pt3), only femoral SUVintegral values decreased at T6, with stable pelvic values. This patient was unable to complete the [$^{18}$F]NaF PET/CT scan on T18, due to severe discomfort. One patient (Pt4) showed a small further increase in femoral head and pelvic SUVintegral values at T6, with return to baseline at T18.

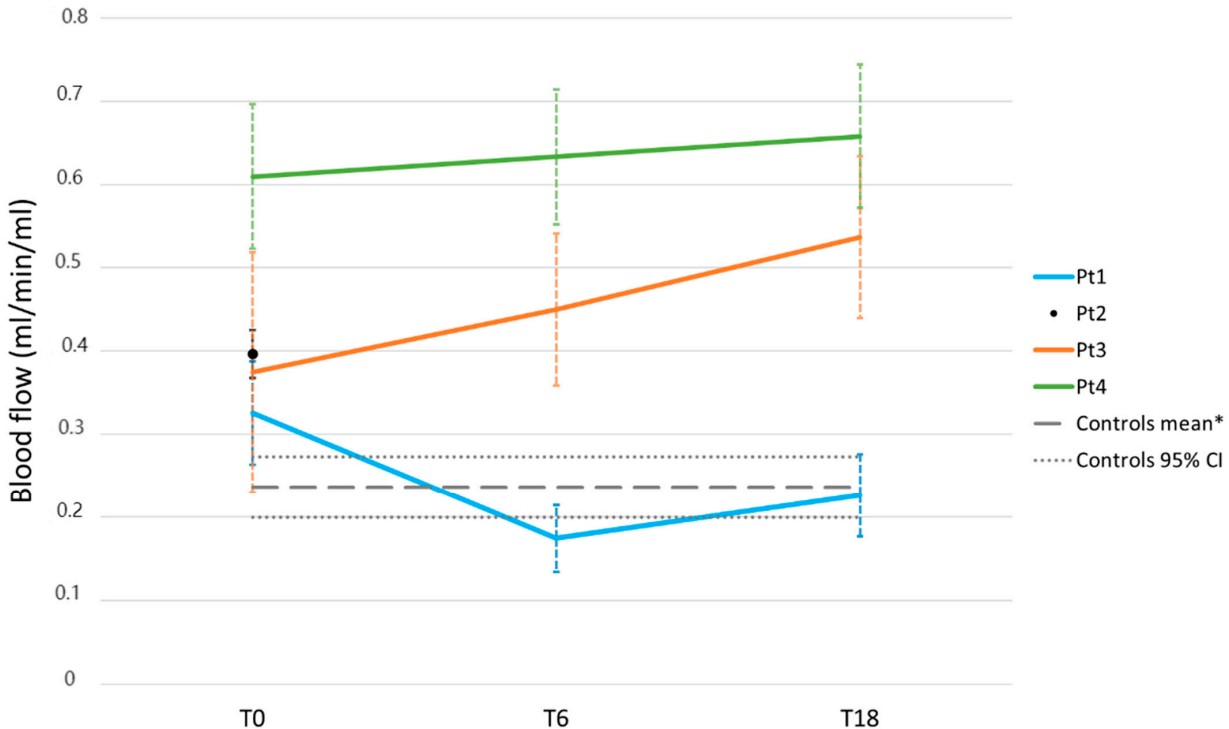

**Figure 5.** Blood flow as measured by [$^{15}$O]water PET. The mean vertebral blood flow (+/−SD) across the three vertebrae (Th5, Th7, and Th9) is represented by the horizontal lines. Vertical dashed lines represent the standard deviation of the mean of the measured values in the three individual vertebrae.* Obviously, healthy controls were examined at baseline only, yielding a mean value across controls with a 95% confidence interval. These results were extrapolated for later time points.

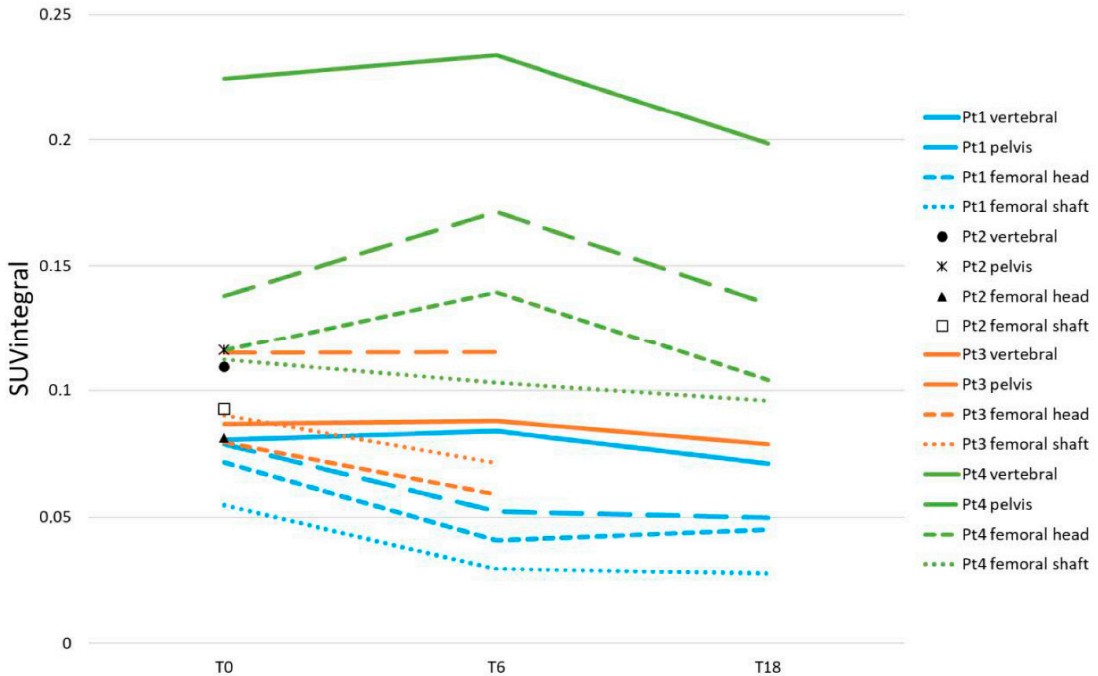

**Figure 6.** SUVintegral as measured by [$^{18}$F]NaF PET.

*3.4. Comparison of Histopathology and Imaging Results*

Histopathological results and imaging results are presented in Tables 3 and 4, respectively. Unless otherwise specified, pelvic imaging results were used for comparisons.

BM fat content was evaluable in five biopsies, and it was low in all. This was in accordance with isointense T1-signals in 4/5 cases, whilst the T1-signal was inhomogeneously hyperintense in the fifth (Pt3 T18). Changes in fat content during the treatment were evaluable in four biopsies. T1-weighted MRI and Dixon reflected the direction of these changes in 2/4 cases and 3/4 cases, respectively. In one case, T1-signals and fat–water fractions were stable despite an observed decrease in BM fat content (Pt3 T18). In one case of increasing BM fat content, fat–water fractions increased but the T1-signal remained isointense (Pt4 T6).

BM cellularity and fibrosis were evaluable in eight biopsies. In 3/4 cases with fibrosis and hypercellularity T1-signals were isointense, whilst the signal was inhomogenously hyperintense in the fourth (Pt3 T18). Of the four cases with fibrosis and low/normal BM cellularity, two showed isointense signals and two showed hyperintense signals. Changes in BM cellularity and fibrosis during treatment were evaluable in four biopsies. No significant changes in fibrosis grade were reported. A decrease in BM cellularity was seen in two cases, which was reflected by an increase in T1-signals and fat–water fractions in 2/2, and by an increase in vertebral $V_e$ in 1/2. In a case with stable BM cellularity, the T1-signal remained unchanged but the fat–water fraction and vertebral $V_e$ increased slightly (Pt4 T6). In one case, T1 signals, fat–water fractions, and vertebral $V_e$ were stable despite an increase in BM cellularity (Pt3 T18).

MVD was evaluable in seven biopsies. Vertebral blood flow as measured by [15O]water PET/CT was increased in all patients with high MVD (at a cut-off value of 0.273 mL/min/mL). Of the three cases with normal MVD, vertebral blood flow was normal in one and increased in two. Changes in MVD during treatment were evaluable in three biopsies. An increase in MVD in one case was reflected by vertebral blood flow as measured by [15O]water PET/CT, but not $K^{trans}$. In one case with persistently increased MVD, [15O]water PET/CT indicated a slight increase in blood flow whilst $K^{trans}$ remained stable. In another case with persistently increased MVD, the opposite occurred.

Osteosclerosis was present in all biopsies, but could be quantified in none, due to fragmentation, crush artefacts, and/or tangentially cut samples.

## 4. Discussion

At baseline, a similar imaging pattern was seen in all patients, consisting of a low BM fat content on T1-weighted MRI/Dixon and increased BM perfusion and osteoblastic activity as measured by [15O]water PET/CT and [18F]NaF PET/CT, respectively. Thus, our combination of imaging techniques seemed to reflect the various BM abnormalities associated with severe MF. Moreover, imaging results provided additional information on disease distribution compared with BM biopsies. In accordance with previous studies in MF [17], our T1-weighted MRI and Dixon results indicate that the BM fat content is the lowest in the vertebrae, sequentially followed by the pelvis and the proximal femora. In addition, we show that abnormal osteoblastic activity as measured by [18F]NaF PET/CT follows a similar distribution pattern.

During ruxolitinib treatment, imaging results revealed differences between patients, which seemed to be congruent with clinical responses. The fastest and most extensive response to all imaging techniques was seen in the patient who first obtained clinical improvement according to conventional response criteria (Pt 1), whilst no significant changes in T1-signal intensities, vertebral blood flow, or [18F]NaF SUVintegral values were seen in the patient with stable disease (Pt 4). Patient 3, who obtained clinical improvement at T18, only showed slight changes in pelvic and/or femoral T1 signal intensities, fat–water fractions, and [18F]NaF SUVintegral values. To our knowledge, we have been the second to prospectively evaluate the imaging appearance of the BM during ruxolitinib treatment. In line with our findings, Luker et al. described heterogeneous response patterns on pelvic/femoral quantitative Dixon MRI amongst four MF patients treated with ruxolitinib [21]. Interestingly, the spleen response—which is often used to monitor therapy effect—was not found to correlate with imaging findings in either of our studies.

Hitherto, we found only one larger prospective study which has used bone marrow imaging in the follow-up monitoring of myelofibrosis patients after treatment. In this study by Sale et al. [17], MRI (T1-weighted and STIR) was performed before and after allogeneic stem cell transplantation, which is currently the only potentially curative treatment in myelofibrosis. They showed a (partial) normalization of T1 and STIR signals in the majority of patients, which started in the femora, and later progressed to the pelvis and vertebrae. Interestingly, we found a similar pattern in our patients treated with ruxolitinib.

Comparing imaging results to histopathological parameters, we found no relevant correlations. However, these comparisons were complicated by several factors. The sample size in our pilot study was small. Moreover, multiple BM biopsies were not evaluable, which highlights the challenges in obtaining assessable samples. Moreover, we suspected one case of BM sampling error—supported by the presence of patchy disease on T1-weighted images (Pt3 T18)—and the comparison of pelvic BM biopsies to vertebral imaging results was suboptimal. These factors, which mainly constitute limitations to the gold standard, have also been reported in a larger study by Sale et al. and might explain why no other studies have reported measures of diagnostic accuracy of our selected imaging techniques in MF [17,21]. Of note, the accuracy of the current gold standard (i.e., pelvic BM biopsies) in early treatment monitoring is further questioned by our T1-weighted MRI and Dixon results, which show that the earliest changes during treatment occur in the proximal femora [17].

As a single technique, we feel that T1-weighted MRI best reflected the response to ruxolitinib therapy. This is in line with a study by Sale et al., which showed that T1-weighted MRI/short-tau inversion recovery (STIR) reflects treatment response after allo SCT in MF [17]. The possibility of quantification of BM fat content makes Dixon an attractive alternative [21]. However, the optimal quantification method is unknown. Fat–water fractions can easily be derived from two-point Dixon images, but they do not distinguish increases in fat content from decreases in water content. Several other methods require additional software [39]. Importantly, although both T1-weighted MRI and Dixon seem to reflect BM fat content, they cannot distinguish fibrosis and hypercellularity. Future studies are needed to evaluate the possible prognostic significance of results [40].

In order to more specifically assess the osteosclerotic component of the disease we used [$^{18}$F]NaF PET/CT. Extending the findings of one available case report [23], we found increased baseline vertebral Ki values in all patients compared to non-MF patients from previous literature [41], presumably reflecting increased osteoblastic activity. Unfortunately, [$^{18}$F]NaF PET/CT results showed no early changes during ruxolitinib treatment in our patients. Nevertheless, given the prognostic relevance of fibrosis and osteosclerosis, [$^{18}$F]NaF PET/CT might be an interesting technique to monitor the effect of future drugs with greater disease-modifying potential. However, it is a time-consuming technique which requires extensive facilities.

In contrast to the above, DCE-MRI seems to be of limited value in the follow-up of MF-related BM abnormalities. For example, increases in $V_e$ were seen both in the patient with a fast clinical response (Pt 1) and in the patient with stable disease (Pt 4). This might be explained by the fact that $V_e$ can be influenced by changes in the number of both hematopoietic cells and fat cells. The interpretation of the parameter $K^{trans}$, which reflects permeability and perfusion [36], proved to be equally challenging. Combining $K^{trans}$ values with [$^{15}$O]water PET/CT results suggested a decrease in BM perfusion and vascular permeability in Pt 1, whilst discrepancies between these parameters implied an isolated decrease in permeability in Pt 3 and an increase hereof in Pt 4. Although these are new findings, with the in vivo effect of ruxolitinib on vascular integrity in MF being largely unknown [42], their reliability is unsure. Previous studies have reported large repeatability values for $K^{trans}$ [43], and it is unknown what constitutes a significant change. Moreover, $K^{trans}$ and $V_e$ are impacted by inaccuracies in native T1 values (see Equation (1)). T1 maps are sensitive to B1 field inhomogeneities, especially at 3 Tesla. We observed systematically high T1 values in patient 2, and a systematic underestimation of T1 values in patient 4

at T18. This might have affected the reliability of $K^{trans}$ and $V_e$ values in these patients. Of note, true changes in T1 due to altered fat ratios can affect $K^{trans}$ and $V_e$ through the same route [44]. Conversely, the parameter $K_{ep}$ is much less sensitive to errors in T1 [45]. However, it does not reflect any specific physiologic parameter and the clinical meaning of changes in $K_{ep}$ during treatment is unclear. We do not exclude a possible prognostic correlation, as has been found in multiple myeloma [46,47].

In light of the above, [$^{15}$O]water PET/CT forms a more reliable alternative for the determination of BM blood flow. Measurements in our healthy controls were fairly constant, albeit slightly higher compared to previous studies in which the lower axial skeleton was evaluated [28,48]. For future studies, we propose the use of a different FOV (i.e., pelvis) for the evaluation of early treatment effects. Moreover, the prognostic significance of changes in BM perfusion should be evaluated. The major limitation for [$^{15}$O]water PET is the short tracer half-life, which requires a production facility on location.

To our knowledge, we have been the first to evaluate BM abnormalities in MF using a multimodal imaging protocol with direct comparisons to BM biopsies. The major limitation of this pilot study was its small sample size. Moreover, the random selection of patients may have led to the exclusion of more frail patients. Larger studies, preferably including patients with various disease stages, are required to validate our results.

## 5. Conclusions

In our pilot study, T1-weighted MRI and Dixon indicated a decreased BM fat content in all patients, with various degrees of reversal during ruxolitinib treatment, starting in the proximal extremities and progressing to the spine. This illustrates the limitations of pelvic BM biopsies for the detection of early treatment responses.

Vertebral BM blood flow as measured by $^{15}$O-water PET/CT was increased in all patients, but ruxolitinib induced a normalization of blood flow in only one patient. Skeletal $^{18}$F-NaF uptake was also increased in all patients and, whilst a minimal decrease was seen in the proximal extremities, vertebral values remained stable during treatment. Diagnostic accuracy was difficult to determine, mainly due to limitations in the gold standard (i.e., the BM biopsy), including a suspected case of sampling error. However, imaging results did seem to correspond to clinical responses during treatment. Moreover, imaging provided additional information on disease distribution throughout the BM. We feel that future research should focus on the prognostic value of imaging results, especially by using whole-body techniques (e.g., T1-weighted MRI, Dixon and [$^{18}$F]NaF PET/CT).

**Author Contributions:** Conceptualization, S.S., C.L., E.B., M.J.W., J.M.Z., S.Z. and P.G.R.; data curation, S.S., C.L. and M.C.H.; formal analysis, S.S., C.L., G.J.C.Z., B.J.H.B., M.C.H., M.Y., E.B. and P.G.R.; funding acquisition, S.Z.; investigation, S.S., C.L., G.J.C.Z., B.J.H.B., J.T.M., M.C.H., J.M.Z. and P.G.R.; methodology, S.S., C.L., G.J.C.Z., B.J.H.B., J.T.M., M.C.H., M.Y., S.Z. and P.G.R.; project administration, S.S.; resources, E.B.; software, C.L., B.J.H.B., J.T.M. and M.C.H.; supervision, M.C.H., M.J.W., J.M.Z., S.Z. and P.G.R.; validation, J.T.M. and M.Y.; visualization, S.S., C.L., G.J.C.Z. and B.J.H.B.; writing—original draft, S.S.; writing—review and editing, S.S., C.L., G.J.C.Z., B.J.H.B., J.T.M., M.C.H., E.B., M.J.W., J.M.Z., S.Z. and P.G.R. All authors have read and agreed to the published version of the manuscript.

**Funding:** Funding was received from Novartis Pharma. Novartis was not involved in the design of the study or in collection, analysis, and interpretation of data. Novartis was also not involved in the writing of, or the decision to publish, the current manuscript.

**Institutional Review Board Statement:** The study was conducted in accordance with the Declaration of Helsinki, and approved by the Central Medical Ethics Committee of VU Medisch centrum (protocol code NL50904.029.14, date of approval 17 September 2014).

**Informed Consent Statement:** Informed consent was obtained from all subjects involved in the study.

**Data Availability Statement:** The datasets used and/or analyzed during the current study are available from the corresponding author on reasonable request.

**Acknowledgments:** The authors thank Matthijs Westerman and Arie van der Spek for the referral of patients. We also thank Otto Hoekstra, Gem Kramer, Adriaan Lammertsma, and Indra Pieters for their input in the early phase of the study design. Lastly, we thank all employees of the Radiology and Nuclear Medicine department who were involved in the scanning of our patients.

**Conflicts of Interest:** M.J.W. and S.Z. received financial support for participation in a steering committee or advisory board of Novartis Pharma. The other authors declare that they have no competing interests.

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
