# Peer review of "Characterizing the Bone Marrow Environment in Advanced-Stage Myelofibrosis during Ruxolitinib Treatment Using PET/CT and MRI: A Pilot Study"

_tomography, doi:10.3390/tomography9020038_

Round 1
Reviewer 1 Report (Previous Reviewer 2)
The authors have made most of their additions and corrections to the discussion, as is appropriate for such a small population studied.
Reviewer 2 Report (Previous Reviewer 1)
The authors have addressed my original comments.
This manuscript is a resubmission of an earlier submission. The following is a list of the peer review reports and author responses from that submission.
Round 1
Reviewer 1 Report
This is an interesting study that uses multimodal imaging to study treatment of myelofibrosis with the drug, ruxolitinib. The manuscript is well written and easy to follow. The main drawback of the study, as the authors point out, is the small size of the study. I have minor suggestions for modifications.
Please work on graphs in figures 2,4,5 to make them easier to read.
Elaborate on the caption in Figure 1. What are the T1 weighted images and what do they show?
How were the error bars determined for the data plotted in figures 4 and 5?
Would you estimate the errors in the imaging data in Table 4? Why do some measurements have three significant figures and others only two?
Reviewer 2 Report
In their manuscript, Slot and Colleagues from an academic medical center in the Netherlands assessed radiographic features of fibrosis as they related to histologyic findings in the setting of a diagnosis of Myelofibrosis prior to, and during treatment. the authors included a "random sample" of myeloproliferative disease patients who had not had current or prior treatment with JACK inhigbitor, and who had intermediate- or high-risk disease according to IPSS stagine of myelofibrosis. What was the selection "random" and how was it determined who would be included? Was there a possibility of selection bias in the choice of patients included for study? Imaging, disease-related data, and histopathology of the marrow were completed prior to treatment, and at month six and month 18 of therapy. Standard risk assessment was done, without the use of molecular pathology. Was this done due to the time period over which patients were enrolled? MRI as well as PET-CT scan assessment tools to assess for the severity of fibrosis were described by the authors, and also included features of osteosclerosis. Were the readers blinded to findings by histopathology? Was there an independent group of readers for both MRI and for PET/CT?
The study population was small, including four patients age 47-75. Unfortunately, patient 2 died before the six-month time interval. A control patient was identified who had had evaluation for herniated disks. Three of four patients had JAK2V617F mutation and all four patients were treated with ruxolitinib. After initiation of therapy, two patients showed an increase in MRI signal intensity, one patient had decrease, and one patient, as stated above, was not assessed. The authors used a variety of other radiographic endpoints including fat-water fractions at different sites, as well as vertebral blood flow, radioactive NaF PET scanning and 15O-water PET. These results were compared to results of histopathology. Again, the number of specimens is small for the small number of patients entered. For example, changes in fat content during treatment were evaluable in only four biopsies. Unfortunately, the authors could not find correlations between imaging and histopathology. Was this due to the small sample size? There were some differences that took place during treatment, based on radiographic findings. how does this compare to what may already have been demonstrated in large trials of ruxolitinib or other drugs identified as active in myelofibrosis? It seems that PET/CT did not prove effective in evaluation of patients, but the study duration was short, and the number of subjects, again was small.
